# A new model for simultaneous dimensionality reduction and time-varying functional connectivity estimation

Diego Vidaurre[1,2,¤]*

**1** Center for Functionally Integrative Neuroscience, Department of Clinical Medicine, Aarhus University, Aarhus, Denmark, **2** Department of Psychiatry, University of Oxford, Oxford, United Kingdom

¤ Current address: Center for Functionally Integrative Neuroscience, Department of Clinical Medicine, Aarhus University, Aarhus, Denmark
* dvidaurre@cfin.au.dk

**Data Availability Statement:** The data is a public repository available online: http://www.humanconnectomeproject.org/data/, upon acceptance of the corresponding license. The data can be downloaded from the website directly in a

## Abstract

An important question in neuroscience is whether or not we can interpret spontaneous variations in the pattern of correlation between brain areas, which we refer to as functional connectivity or FC, as an index of dynamic neuronal communication in fMRI. That is, can we measure time-varying FC reliably? And, if so, can FC reflect information transfer between brain regions at relatively fast-time scales? Answering these questions in practice requires dealing with the statistical challenge of having high-dimensional data and a comparatively lower number of time points or volumes. A common strategy is to use PCA to reduce the dimensionality of the data, and then apply some model, such as the hidden Markov model (HMM) or a mixture model of Gaussian distributions, to find a set of distinct FC patterns or states. The distinct spatial properties of these FC states together with the time-resolved switching between them offer a flexible description of time-varying FC. In this work, I show that in this context PCA can suffer from systematic biases and loss of sensitivity for the purposes of finding time-varying FC. To get around these issues, I propose a novel variety of the HMM, named HMM-PCA, where the states are themselves PCA decompositions. Since PCA is based on the data covariance, the state-specific PCA decompositions reflect distinct patterns of FC. I show, theoretically and empirically, that fusing dimensionality reduction and time-varying FC estimation in one single step can avoid these problems and outperform alternative approaches, facilitating the quantification of transient communication in the brain.

## Author summary

I show that PCA, although widely used in practice, can introduce important biases and loss of sensitivity in the estimation of time-varying functional connectivity on high-dimensional fMRI data. I discuss these limitations and propose a new method that, by performing dimensionality reduction and time-varying functional connectivity estimation in one single step, can effectively overcome these limitations.

browser or through an Amazon Simple Storage Solution (S3) bucket. Alternatively, a hard drive with the data can be ordered from the website. Code availability The toolbox supporting all the HMM variants (and mixture of models) compared in this paper can be found in http://https://github.com/OHBA-analysis/HMM-MAR. User documentation for these and other HMM variants can be consulted in https://github.com/OHBA-analysis/ HMM-MAR/wiki. The scripts to run all examples on synthetic or real data can be found in https://github.com/OHBA-analysis/HMM-MAR/blob/master/examples/PLOSCompBio.m.

**Funding:** The study was supported by a Novo Nordisk Foundation Emerging Investigator Fellowship (NNF19OC-0054895) and an ERC Starting Grant (ERC-StG-2019-850404) to DV. The funders had no role in study design, data collection and analysis, decision to publish, or preparation of the manuscript.

This is a *PLOS Computational Biology* Methods paper.

## Introduction

When we image the brain of passive subjects with fMRI, the measured signals exhibit strong correlations even between areas that are far apart in the brain [1, 2]. These patterns of resting-state correlation, referred to as functional connectivity (FC), are interpreted as a sign that these regions are somehow engaged together in relation to one or more brain processes [3]. It is now widely recognised that FC holds important relations to mental and clinical phenotypes, is reliably subject-specific (i.e. reproducible across scanning sessions), and is also hereditary [4–6]. However, the mere existence and interpretability of within-session modulations in FC is, at least in fMRI, more controversial; see [7, 8] for arguments in both directions. An important reason for this dispute is the scarcity of time samples and the very high dimensionality of the data. In this context, quantifying modulations of FC within session is a challenging statistical problem because these changes (if they exist) are by definition spontaneous and have no obvious behavioural reference [9].

One possible strategy to quantify time-varying FC is the use of sliding-windows, where some measure of FC such as Pearson's correlation is computed across regions for each window in the data, typically followed by the application of a clustering algorithm to extract patterns across windows [10]. Although attractive because of its simplicity, this method suffers from an important problem of statistical variability in the estimation, such that disentangling actual changes in the data from fluctuations caused by statistical noise is not straightforward [11]. Alternatives to Pearson's correlation include phase coherence [12, 13], and the angle [14] or covariance [15] between the signal gradients; these have been shown to necessitate shorter windows but might however be more sensitive to high-frequency noise artefacts.

Dispensing with the use of windows, methods that boost the statistical power by using the entire data set in the estimation are sometimes preferred. One such method is the Hidden Markov model (HMM), which assumes that the data can be reasonably modelled using a discrete number of FC states with Markovian dynamics [16]. For example, if an HMM with twelve states was trained on 820 subjects from the Human Connectome Project (HCP) data set [17], each state would be on average estimated on 68.3h of data; compared to a typical 1min window, the statistical noise in this estimation is very small. In the HMM, the Markovian property means that the model accounts for the state dynamics using a probability matrix that encodes the probability of transitioning between each pair of states—but without modelling the previous history of state activations [18, 19]. A straightforward variant of the HMM is to model each state as a Gaussian distribution where the mean is pinned to zero in order to prioritise changes in covariance [8]. An alternative is the mixture model of Gaussian distributions [19], which has no transition probability matrix and therefore ignores the temporal structure of the data.

Unfortunately, neither the HMM nor the mixture model of Gaussian distributions are easily applicable when the data dimensionality (the number of voxels) is too high in relation with the number of time points (volumes). The two most common approaches for reducing dimensionality in this context are using an anatomical parcellation [20] and applying independent component analysis (ICA) [3]. These produce a number $n$ of regions or components (typically a hundred or more), so that an FC matrix has $n(n-1)/2$ different parameters. Often, this is still high enough for the HMM inference (or mixture model inference) to overfit and produce degenerate solutions in many data sets. For this reason, PCA is usually carried out on the

ICA-derived or parcellated time series so that the HMM (or alternative method) is run on an even lower-dimensional space [21].

Whereas this two-step method works reasonably well in practice for many (but not all) fMRI data sets, in the sense that it captures within-session FC modulations at least to some extent, having PCA and the HMM estimation as two separate steps is suboptimal. This is because the PCA step specialises in maximising explained variance, and is not designed to the specific goal of quantifying within-session FC modulations. Furthermore, the use of PCA can unknowingly introduce important biases on the estimation. In this paper, I discuss these issues in detail and propose an alternative model that bypasses these problems: an HMM where each state is a PCA decomposition. I refer this model to as *HMM-PCA*. Critically, because the computation of PCA is based on the data covariance [22], the state-specific PCA decompositions reflect distinct patterns of FC, effectively fusing dimensionality reduction and time-varying FC estimation in one single step. Fig 1 presents a graphical illustration of both the HMM with Gaussian states over principal components (HMM-Gaussian, top), and the HMM-PCA (bottom).

## Materials and methods

### The problem of estimating time-varying FC in high dimensions

Let $d_t \in \mathbb{R}^{1 \times n}$ be the multivariate source signal at volume (time point) $t = 1 \ldots T$, so that $\boldsymbol{D}$ denotes the data concatenated for all sessions and subjects –although this can also be applied at the single-subject level provided that we have a sufficient number of volumes. Here, $n$ corresponds to anatomical parcels or ICA components, referred to generically as regions. A standard estimation of functional connectivity (FC) for subject $j$ is an $n \times n$ matrix $\mathbf{C}$ containing the Pearson's correlation coefficient for each pair of regions. Formally, the question at hand can be posed as: can we find differences between matrices $\mathbf{C}(t_1)$ and $\mathbf{C}(t_2)$, defined as *instantaneous* FC matrices at time points $t_1$ and $t_2$, for at least one pair of time points $t_1$ and $t_2$ belonging to the same scanning session?

One way to approach this problem is the use of the Hidden Markov Model (HMM), which describes the data in terms of a finite number $K$ of states that activate or deactivate throughout the scanning time. The state time courses, reflecting these activations, are in the form of probabilities $P(x_{tk} = 1|\boldsymbol{D}) = \gamma_{tk}$, where $x_{tk} = 1$ means that the state $k$ is active at time point $t$, and $\gamma_{tk}$ is the estimated posterior probability of the event $x_{tk} = 1$, given the data. The HMM is a generic model where the states can be described using any family of probability distributions. Within this general framework, we can define the states as covariance matrices $\Sigma_k$. In this case, the states are characterised as Wishart distributions or, equivalently, as zero-mean Gaussian distributions [8]. This means that, when the $k$-th state is active, the data is considered to be generated according to the distribution

$$d_t \sim \mathcal{N}(\mathbf{0}, \Sigma_k),  \tag{1}$$

I refer to this approach as *HMM-Gaussian*. Alternatively, the mixture model of Gaussian distributions dispenses with the transition probability matrix, thus ignoring the temporal structure of the data and treating the time points (volumes) as independently distributed and exchangeable [19]. In this case, the states are also defined as Wishart or zero-mean Gaussian distributions, but the transition probability between consecutive time points is not modelled. I refer to this model as *Mix-Gaussian*. In either case, an instantaneous FC matrix at time point $t$ can be defined as a linear combination of the state covariance matrices $\Sigma_k$ using the state probabilities $\gamma_{tk}$ as weights.

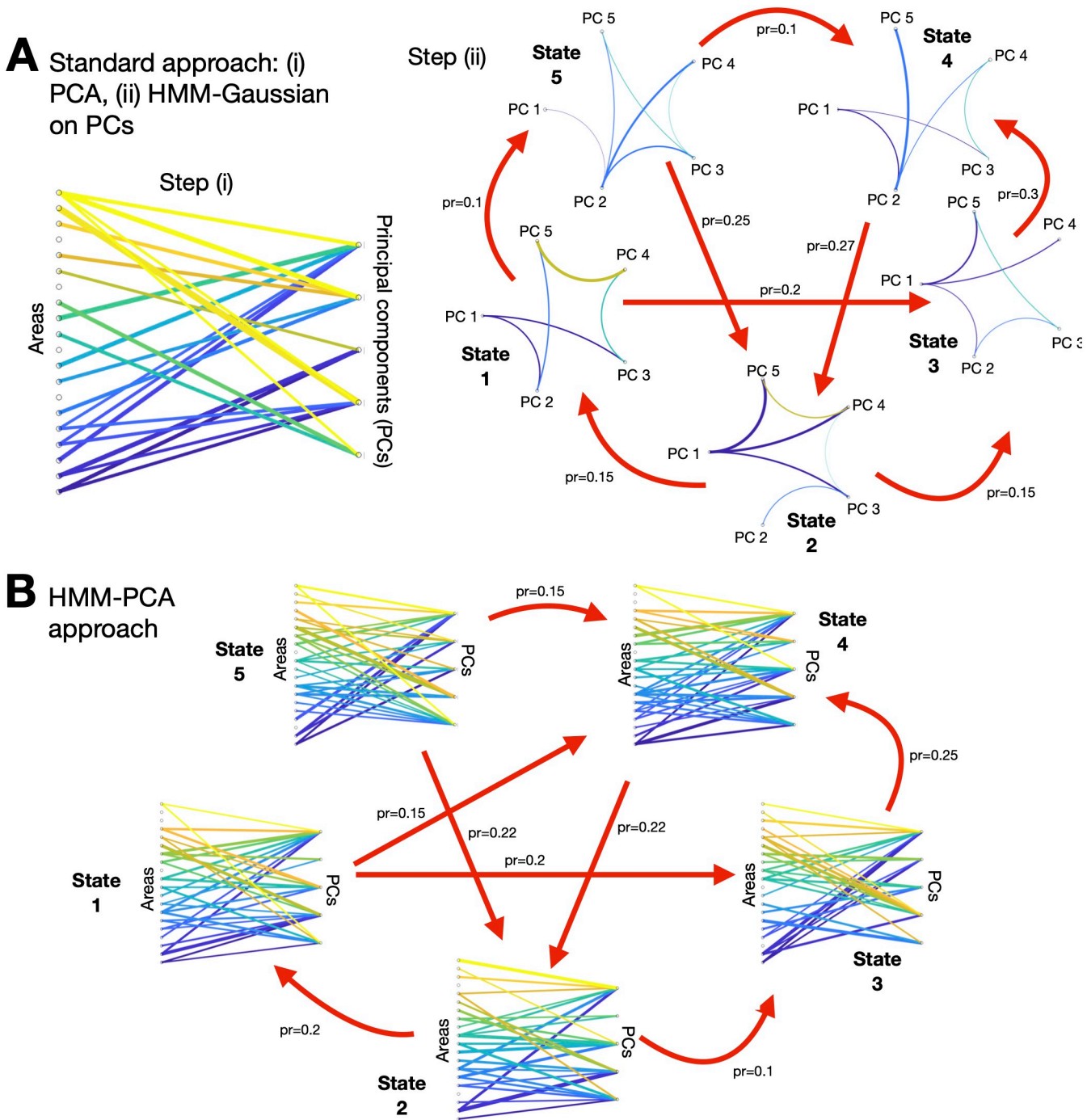

**Fig 1. Two different approaches for the estimation of time-varying FC on high-dimensional fMRI data. A**. PCA is first used as a dimensionality reduction step, blindly to the purpose of estimating time-varying FC; then, some state-based model (like the hidden Markov model) is run on the first principal components (PC). **B**. The HMM-PCA approach, where each state is a different PCA decomposition, is run directly on the high-dimensional data; given that the computation of PCA is based on the data covariance, different PCA decompositions capture different patterns of FC. See S1 Fig for representations in the form of graphical models.

Both the HMM and the mixture model necessitate the number of states $K$ to be pre-specified. Although there are extensions of the models where the number of states is directly inferred from the data (referred to as infinite HMM [23] or infinite mixture model [24], respectively), their parameter estimation is computationally demanding when we have large amounts of data and, in fMRI, they often yield similar results to the "finite" models [25]. Therefore, here we prespecify $K$ and submit the data to an inference algorithm that will return the state probabilities $\gamma_{tk}$, the state parameters ($\Sigma_k$), and the transition probability matrix and $\gamma_{tk}$.

Because $n$ is often large in comparison to $T$, PCA is typically used as an intermediate dimensionality reduction step. This way, HMM-Gaussian (or Mix-Gaussian) is estimated on $Y_t = d_t W$, where $W \in \mathbb{R}^{n \times p}$ represents a PCA decomposition and $p$ is the number of principal components (PCs). The estimated FC matrices are thus low-dimensional, $\tilde{\mathbf{C}}(t) = W' \mathbf{C}(t) W$. Note that, across the entire PCA-reduced data set, $Y \in \mathbb{R}^{T \times p}$, the $p$ PCs are by construction orthogonal (i.e. the correlation between the columns of $Y$ is zero). However, there might be periods in the data during which the time series are temporarily not orthogonal, meaning that the time series are temporarily correlated (or negatively correlated) across brain regions. Any transient departure from (the time-averaged) orthogonality will be encoded by the HMM parameters $\gamma_{tk}$ and $\Sigma_k$, and can be considered an FC modulation.

## HMM-PCA: A new model for estimating time-varying FC

I now introduce the HMM-PCA mathematically. Various of the elements of this model are analogous to the probabilistic mixture model of PCA analysers introduced by Tipping and Bishop [26] (here referred to as *Mix-PCA*), which, similarly to Mix-Gaussian, does not account for the temporal structure of the fMRI data. The model is also closely related to that of Alvárez and Henao [27]. In brief, the main read-outs of HMM-PCA and Mix-PCA are (i) a set of $K$ states, each characterised by a PCA decomposition; and (ii) the corresponding state time courses $\gamma_{tk}$, which encode the probability of each state $k$ to be active at each time point $t$. In the case of HMM-PCA, a matrix with transition probabilities between states is also estimated.

Both Mix-PCA and the HMM-PCA are based on probabilistic PCA [28], which formulates classic PCA within a probabilistic framework. Probabilistic PCA assumes the following distribution:

$$d_t | y_t, \boldsymbol{\mu}, W, \sigma^2 \sim \mathcal{N}(\boldsymbol{\mu} + y_t W', C), \tag{2}$$

where the covariance is given by $C = WW' + \sigma^2 I$, and $I$ is the identity matrix. The density function is therefore

$$P(d_t | y_t, \boldsymbol{\mu}, W, \sigma^2) = (2\pi\sigma^2)^{-n/2} \exp\left(-\frac{1}{2\sigma^2} \|d_t - y_t W' - \boldsymbol{\mu}\|^2\right), \tag{3}$$

where $y_t \in \mathbb{R}^{1 \times p}$ is assumed to be Gaussian distributed, and $\|\cdot\|$ is the Euclidean norm. Since for our purposes we are not interested in modelling transient changes in amplitude, and we wish to concentrate the model's explanatory power on FC modulations as much as possible, I introduce the modification $\boldsymbol{\mu} = \mathbf{0}$ to the model proposed by [26], therefore not modelling changes in amplitude explicitly.

To model the data, the Mix-PCA model uses, as states, $K$ different PCA projections, $W_k \in \mathbb{R}^{n \times p}$, and their corresponding noise variance estimations $\sigma_k^2$, which are estimated from the data together with the state occupancies. The state covariance matrices are denoted as $C_k = W_k W_k' + \sigma_k^2 I$. The prior probability for each state to have generated unseen data is given by $\pi_k$, which also needs to be estimated. Therefore a second set of latent variables $x_k =$

$(x_{1k}, ..., x_{Tk}) \in \mathbb{R}^{1 \times K}$ is required, such that $x_{tk} = 1$ if the $k$-th component (or state) is responsible for having generated the observed data at time point $t$, and $x_{tk} = 0$ otherwise. I define $\boldsymbol{X} = [\boldsymbol{x}_1 ... \boldsymbol{x}_K]$, and refer to the posterior probabilities $P(x_{tk} = 1|\boldsymbol{d}_t, \pi_k) = \gamma_{tk}$ as the state time courses. Note that $\boldsymbol{x}_t$ is conditionally independent of all data time points except $\boldsymbol{d}_t$, therefore ignoring the data temporal structure. See S1 Fig for a graphical representation of the model conditional independences of HMM-PCA and the other variants. The posterior probabilities $\gamma_{tk}$ can be estimated, for example, using the expectation-maximisation (EM) algorithm [26, 28], or which I provide some details below.

In the HMM-PCA case, I instead have the state latent variables $\boldsymbol{x}_k$ modelled as an order-1 Markovian process, with prior probabilities $P(x_{tk_1} = 1|x_{(t-1)k_2} = 1) = \Theta_{k_2 k_1}$. Here, $\boldsymbol{\Theta}$ is the transition probability matrix, which models the average probabilities of transitioning from one state to another—and must be estimated as well. This way, the posterior probabilities (namely, the state time courses) are defined as $P(x_{tk} = 1|x_{(t-1)}, \boldsymbol{d}_t, \boldsymbol{\Theta}) = \gamma_{tk}$.

To use the EM algorithm to solve the HMM-PCA problem (and assuming for simplicity of notation that we have one single, continuous time series), the expected log-likelihood can be formulated as

$$
\begin{aligned}
\langle \log \mathcal{L} \rangle = \Bigg( \sum_{t=1}^{T} \sum_{k=1}^{K} \gamma_{tk} \Bigg( &-\frac{n}{2} \log(2\pi) - \frac{n}{2} \log \sigma_k^2 - \frac{1}{2} \mathrm{trace} \langle \boldsymbol{y}_t' \boldsymbol{y}_t \rangle \\
&- \frac{1}{2\sigma_k^2} \boldsymbol{d}_t \boldsymbol{d}' + \frac{1}{\sigma_k^2} \langle \boldsymbol{y}_t \rangle \boldsymbol{W}_k' \boldsymbol{d}_t' \\
&- \frac{1}{2\sigma_k^2} \mathrm{trace} \big( \boldsymbol{W}_k' \boldsymbol{W}_k \langle \boldsymbol{y}_t' \boldsymbol{y}_t \rangle \big) \Bigg) \Bigg) + \\
&\langle \log P(x_0) \rangle + \sum_{t=2}^{T} \langle \log P(x_t, x_{t-1}), \rangle,
\end{aligned}
\tag{4}
$$

where $\langle \cdot \rangle$ denotes expectation and $(\boldsymbol{W}, \sigma^2)$ comprise these variables for $k = 1...K$, and where

$$
\begin{aligned}
\langle \boldsymbol{y} \rangle &= \boldsymbol{M}_k^{-1} \boldsymbol{W}_k' \boldsymbol{d}_t' \\
\langle \boldsymbol{y}_t' \boldsymbol{y}_t \rangle &= \sigma_k^2 \boldsymbol{M}_k^{-1} + \boldsymbol{y}_t' \boldsymbol{y}_t \\
\boldsymbol{M}_k &= \sigma_k^2 I + \boldsymbol{W}_k' \boldsymbol{W}_k.
\end{aligned}
$$

Similarly to the Mix-PCA model, the EM updates for $\boldsymbol{W}_k$ and $\sigma_k^2$ are coupled. Using the same estimation of the intermediate variable $\boldsymbol{M}_k$ for both, the new parameter estimates for these variables become:

$$
\begin{aligned}
\hat{\boldsymbol{W}}_k &= S_k \boldsymbol{W}_k (\sigma_k^2 \boldsymbol{I} + \boldsymbol{M}_k^{-1} \boldsymbol{W}_k' S_k \boldsymbol{W}_k)^{-1} \\
\hat{\sigma}_k^2 &= \frac{1}{n} \mathrm{trace} \big( \boldsymbol{S}_k - \boldsymbol{S}_k \boldsymbol{W}_k \boldsymbol{M}_k^{-1} \hat{\boldsymbol{W}}_k \big),
\end{aligned}
\tag{5}
$$

where

$$
S_k = \frac{1}{\sum_{t=1}^{T} \gamma_{tk}} \sum_{t=1}^{T} \gamma_{tk} \boldsymbol{d}_t' \boldsymbol{d}_t
$$

is the state-specific (sample) covariance matrix in the original space, such that the aggregated sample covariance matrix $\boldsymbol{S} = (1/T) \sum_{t=1}^{T} \boldsymbol{d}_t' \boldsymbol{d}_t$ can be expressed as a weighted average of $S_k$.

For the full EM algorithm, it only remains the estimation of the state activation probabilities $\gamma_{tk}$, the transition probability matrix $\Theta$, and the initial probabilities $\pi$ for the first time point of the scans.

As is standard practice, the $\gamma_{tk}$ parameters are estimated using the forward-backward equations, given the likelihood for each HMM state (here, PCA decomposition) at $t$ [18]. The update rules for $\Theta$ and $\pi$ are also equivalent to any other HMM (given the expected log-likelihood in 5), and can be found elsewhere [19, 29].

## Results

### Limitations of the two-step approaches

Previous work has shown that the HMM, when ran on PCA time series, can produce useful representations of the data [21]. However, this approach suffers from two limitations: (i) a loss of statistical efficiency in detecting time-varying FC when there is time-varying-FC-relevant information in the discarded PCs, and (ii) a bias towards the lower-order PCs that were included in the model (that is, those explaining less variance). These limitations, which I discuss next, equally apply to other probabilistic models and clustering methods such as Mix-Gaussian when applied on PCA time series. Note that the following discussion merely states two mathematical facts intrinsic to the use of PCA. How much these biases actually affect real data will likely depend on many factors including data acquisition, preprocessing, and experimental paradigm.

**Loss of sensitivity.**   I first show the loss of sensitivity in detecting time-varying FC when the discarded PCA components contain time-varying-FC-related information. Given that within-session modulations in FC are bound to be subtle [7, 8], it is quite possible that such modulations will indeed occur in lower-order PCs. Since this is not straightforward to show without access to the ground truth, I used a simulation where, as it happens in fMRI, the temporal modulations of covariance are not very large. This simulation illustrates that HMM-PCA may outperforms competing models if there is time-varying FC in lower-order PCs. Secondarily, these simulations stress the importance of acknowledging the temporal nature of the data, which is ignored by Mix-PCA.

I generated data from two different simulation schemes. In both cases, the data were generated from a low-dimensional space and projected to the dimension of the observed data ($n = 10$), where some observational 10-dimensional white noise was added. (That is, the data is low-rank up to the observational noise). The nature of this projection varies according to two different states, which transitions are organised as an order-1 Markovian process with transition probabilities, $P(x_{tk_1} = 1 | x_{(t-1)k_2}) = .9615$ if $k_1 = k_2$ and $P(x_{tk_1} = 1 | x_{(t-1)k_2}) = .0385$ otherwise; that is, the probability of remaining in the same state is 25 times higher than that of transitioning. I sampled 10 sessions of 1000 data points each. Note that I generated the data from a Markovian process to make it simple, but the fact that the generative process aligns with the Markovian assumption of the HMM does not result in any loss of generalisation for the main point of the simulations.

In Scenario 1, I separately sampled each of the latent dimensions from a zero-mean Gaussian distribution, so that these are approximately orthogonal. I denote the dimension of the latent space as $p_0$, and the generated latent data as $Y_0$. Two different separate cases are considered: in the first, I have $p_0 = 2$, where I sampled two latent variables with standard deviations 2.0 and 1.5; in the second, I have $p_0 = 3$, with respective standard deviations of 2.0, 1.5 and 1.0. I then sampled three random (standard Gaussian-distributed) projection matrices $A, A_1, A_2 \in \mathbb{R}^{n \times p_0}$, and generated the observed data as $S = Y_0(A + A_1) + \epsilon$ when state 1 is active, and $S = Y_0(A + A_2) + \epsilon$ when state 2 is active. Observational noise $\epsilon$ is set to have zero mean

and standard deviation equal to 0.001. The ground-truth state time courses (telling when each state is active) were generated using Markov chain Monte Carlo sampling as mentioned above.

In Scenario 2, I based the sampling on actual fMRI data from the HCP (see above). In particular, I randomly chose $n = 10$ regions from the data and computed the data covariance matrix $C$, which I then eigendecomposed into its principal components: $C = V'EV$, where $E$ is a diagonal matrix of eigenvalues and $V$ contains the corresponding eigenvectors. I used these to create two covariance matrices $C_1$ and $C_2$, which were then used to sample the data for each of the two states given a multivariate Gaussian distribution with zero mean. As with the other scenario, I considered two cases: $p_0 = 2$ and $p_0 = 3$. In either case, the first eigenvector of both $C_1$ and $C_2$ was set to be the first eigenvector of $C$, thus corresponding to the time-invariant component of FC. In the $p_0 = 2$ case, the second eigenvector of $C_1$ and $C_2$ were each assigned a different permutation of the second eigenvector of $C$; the rest of eigenvectors were set to zero. In the $p_0 = 3$ case, the second and third eigenvectors of $C_1$ and $C_2$ were assigned different permutations of the second and third eigenvector of $C$, and the rest of eigenvectors were set to zero. These latter non-zero eigenvectors, therefore, represent the time-varying components of the FC. Again, additive observational noise $\epsilon$ was set to have zero mean and standard deviation equal to 0.001. The transition probability matrix was designed as before.

I repeated the simulations 50 times, and estimated HMM-PCA, Mix-PCA and HMM-Gaussian models, which were all set to use $p = 2$ components, and, for simplicity, the right number of states $K = 2$. Therefore, there is loss of information only in the $p_0 = 3$ cases. For illustration, Fig 2A shows one specific instance for Simulation 1 and $p = p_0 = 2$, where the HMM-Gaussian

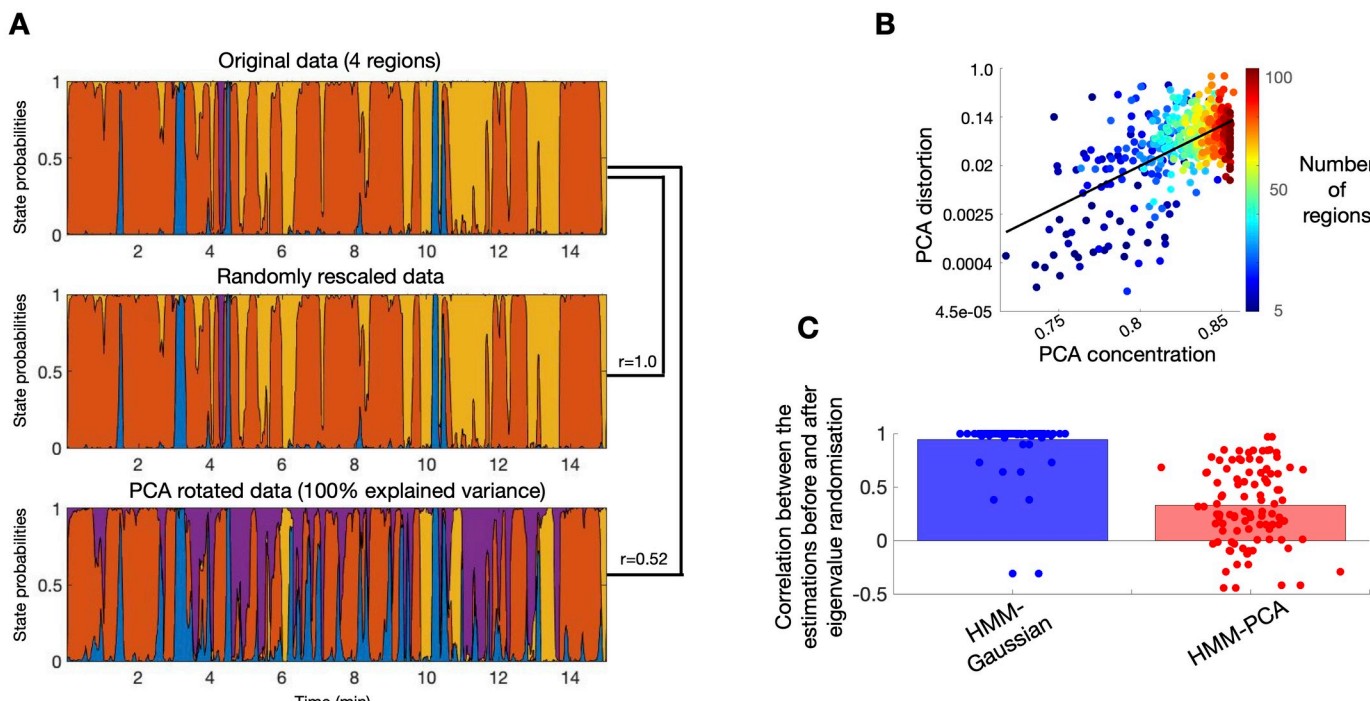

**Fig 2. HMM-PCA outperforms the HMM-Gaussian and Mix-PCA approaches on synthetic data. A.** Example of how the different models recover the ground-truth state time courses. **B.** Comparative accuracy between HMM-PCA (Y-axis), HMM-Gaussian (X-axis, red) and Mix-PCA (X-axis, blue). Each dot represents one repetition of the simulations. Accuracy is measured in terms of how well each method recovered the ground-truth state time courses. Permutation-based statistical testing revealed that HMM-PCA was always significantly more accurate than the other approaches (p<0.001).

approach misses some of the swiftest state changes, and Mix-PCA, on the other hand, appears to be noisier due to the lack of consideration for the temporal structure of the data. Fig 2B shows the complete results. Here, accuracy corresponds to the proportion of time where the corrected state was guessed. Each dot corresponds to one instance of the simulations, representing how the HMM-PCA compares to the Mix-PCA (blue) or to the HMM-Gaussian (red). Therefore, the points that lie to the left of the diagonal line correspond to simulations where the HMM-PCA performed better than the other models, and the points that lie to the right represent simulations where the HMM-PCA performed worse. Given that $K = 2$, and because the order of the states is non-identifiable, this measure of accuracy ranges between 0.5 and 1.0. Models with accuracy of around 0.5 correspond to degenerate solutions, where one of the two states was obliterated by the inference. A summary of the average accuracy is shown in the text boxes. As expected, HMM-PCA has almost perfect accuracy for the $p_0 = 2$ cases, and deteriorates to a moderate extent when $p_0 = 3$. Most importantly, HMM-PCA also outperforms HMM-Gaussian in most simulations, confirming that approaching the problem in one single step leads to superior solutions when there are time-varying FC information in low-order PCs; i.e. when time-varying FC modulations are subtle, as it is the case in most fMRI data sets. HMM-PCA also performs better than the Mix-PCA, highlighting the importance of accounting for the temporal structure of the data. (Note that in real data the temporal structure is stronger than in these simulations, so this difference will be even larger).

**Bias towards low-order PCA components.**   The previous section discussed the suboptimality of the HMM-Gaussian solutions when there is time-varying FC information in lower-order PCs. I now discussed an intrinsic bias that will manifest regardless of how the time-varying FC information distributes across the PCs, and that will occur above and beyond the natural loss of information of PCA. This is that the application of PCA systematically alters the HMM or mixture model inference with regard to the original estimation (i.e. the one obtained without using PCA), biasing it towards low-order PCs and introducing a factor of arbitrariness in the inference. Critically, this issue occurs even when we keep all PCs and retain 100% of the variance.

When states are described as Gaussian distributions, the HMM (or mixture model) inference is scale-invariant. That is, as far as we use the same random seed in the initialisation of the inference, the estimation of the state time courses $\gamma$ will not be affected if we multiply the time series of any given region by any random scalar. Mathematically, this can be expressed as

$$\gamma = G(\boldsymbol{D}; \Upsilon) \approx G(\boldsymbol{D} \operatorname{diag}(\boldsymbol{w}); \Upsilon), \quad \forall \boldsymbol{w} \in \mathbb{R}^n, \tag{8}$$

where $G(\boldsymbol{D}; \Upsilon)$ represents the HMM inference process given some specification of hyperparameters $\Upsilon$ (e.g. the number of states). That is, the inference remains unaltered after rescaling the regions' time series by any vector $\boldsymbol{w}$, regardless of the specific values of such vector. Intuitively, the reason is because the state-specific covariance matrices $\Sigma_k$ can adjust their diagonal (i.e. their variance) to compensate for this global scaling with no effects in the inference. (This is as far as the prior distribution of the covariance matrices acknowledges this scaling; if not, there might be small changes in the inference but rarely substantial provided that we have enough data).

However, the HMM (or mixture model) inference is not rotation-invariant, and, in particular, it is not invariant to a PCA rotation:

$$G(\boldsymbol{D}; \Upsilon) \neq G(\boldsymbol{D}\boldsymbol{W}; \Upsilon). \tag{9}$$

When we apply PCA on the data, $\boldsymbol{Y} = \boldsymbol{D}\boldsymbol{W}$, the columns of $\boldsymbol{Y}$ are ordered according to its variance, such that the first column of $\boldsymbol{Y}$ (the first eigenvector) has the highest variance and the

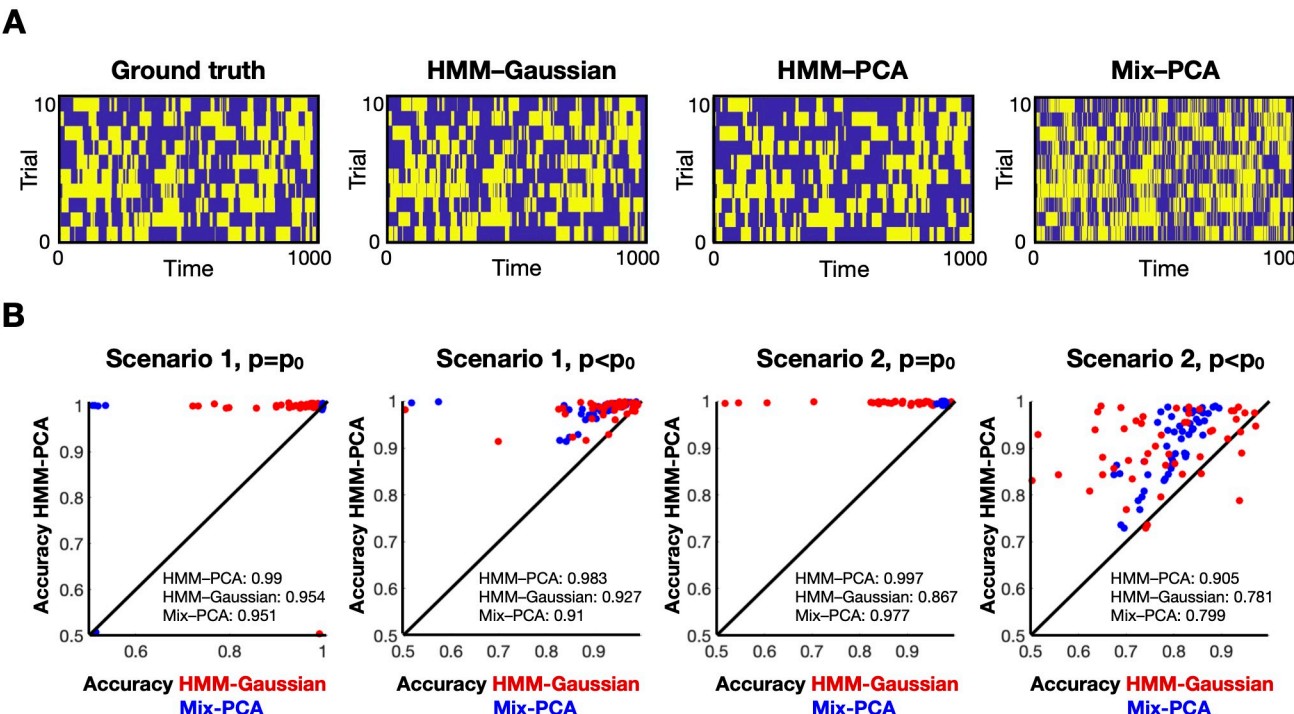

**Fig 3. PCA introduces an estimation bias on the HMM or mixture model. A**. The HMM inference is scale-invariant, but not PCA-rotation-invariant. State time courses produced by the HMM inference for one HCP subject on the original parcellation space (top), after applying a random scaling of the data (middle), and after PCA rotation with no loss of variance (bottom). Each colour represents a different state, so that the coloured areas indicate the probability of activation for the states across the session. The similarity between the different runs, expressed as Pearson's correlation coefficients, are expressed on the right. **B**. The extent of this bias (PCA distortion) is logarithmically related to how concentrated is the variance on the first PCs (PCA concentration); that is, the more correlated are the regions on the original data, the stronger will be the bias introduced by PCA. **C**. Random manipulations of the data eigenvalues are correctly reflected as changes in the HMM-PCA estimations; HMM-Gaussian is not able to capture the changes.

last column has the lowest variance. Because of the scale-invariance property of the HMM inference, however, these variances will be ignored. Intuitively, this means that the low-order PCs (which explain less variance in the original data) are given in principle the same weight in the inference than the high-order PCs (which explain more variance in the original data). In different words, a PCA decomposition has information in both the eigenvectors and the eigenvalues, but the eigenvalues are ignored in the HMM estimation, therefore changing the HMM estimation with regard to what would be obtained in the original data. In practice, that results in a distortion of the estimates with respect to the original data, which will become more drastic as we include more and more low-order PCs.

For example, let us consider one given subject from the HCP data set [17], eight randomly-chosen brain parcels from the (100 regions) Schaefer parcellation [30], and a fixed initialisation of the algorithm (i.e. initialising the inference with exactly the same starting state time courses, so there is no randomness anywhere throughout the inference). Fig 3A shows the estimated state time courses for the original data (top), the state time courses estimated after scaling each channel randomly (middle), and the state time courses obtained from a PCA decomposition where we kept all components so that there is no loss of information (bottom). As observed, PCA-rotating the data changes the estimation, whereas scaling does not.

Furthermore, this bias is related with the number of regions and how concentrated is the variance on the first PCs. This is shown in Fig 3B. The amount of *PCA distortion* (Y-axis) is here quantified as one minus the correlation between the state time courses obtained on the original data vs. those obtained on the PCA projection. A measure of *PCA concentration*

(X-axis) is given by the average cumulative explained variance across PCs; for example, if the areas were perfectly correlated then the first PC would explain all the variance (1.0) and the cumulative explained variance of all PC would be 1.0, in which case the average –i.e. the PCA concentration– would be exactly 1.0; in the opposite case, if the regions were orthogonal (uncorrelated), then the cumulative explained variance of the $j$-th PC becomes $j/n$, and the average becomes exactly 0.5. In each run, I sampled a number (between 5 and 100) of regions from the Schaefer parcellation and run the HMM on both the original and the PCA-projected data (with no loss of variance). Fig 3B shows that there is a logarithmic relation between PCA concentration and PCA distortion across HMM runs, suggesting that the more correlated the regions are, the stronger is the bias introduced by PCA.

It follows the question of whether HMM-PCA has this problem. It can be shown theoretically that HMM-PCA does not have this issue, because, as discussed, the problem has to do with having state-specific error variance parameters, while in our HMM-PCA formulation the error covariance matrix (and therefore error variance) is common to all states. As we have done empirically, we can manipulate the data by randomising their eigenvalues (i.e. by multiplying the ordinary PCA weights by a random number). Since this is changing the nature of the data, the HMM estimation, if correct, should also change to reflect the manipulation. If it does not change, that would signal a problem. Fig 3C shows the correlation between the HMM estimation (i.e. between the state time courses) before and after performing this manipulation in the data, for both HMM-Gaussian and HMM-PCA. As shown, HMM-PCA reflects that the data have changed by changing its estimates, while HMM-Gaussian is mostly unaffected. Although this can only be considered as indirect evidence, it adds further evidence to the issue discussed in this section.

In summary, although PCA is often an acceptable approximation in practice, it can also arbitrarily distort the time-varying FC estimates (with respect to a non-PCA estimation) towards the lower-order PCs. This effect will be more pronounced when the regions are more correlated –i.e. when the proportion of variance explained by the different PCA components is less equally balanced.

## Empirical results

Next, I demonstrate the comparative performance of the models on both real data from the HCP and synthetic data constructed by using aspects of real HCP data –see [3] and references therein for details about the HCP data acquisition and preprocessing.

**Simulated data experiments.** In the previous section, I have used examples to illustrate the limitations of PCA when used in combination with the HMM or similar models. I now explore how the different models behave in higher-dimensional, more realistic data. Although these data are synthetic, I used aspects of real HCP data to perform the simulations.

Specifically, using the 100-regions Schaefer parcellation [30] and taking a standardised time series for each parcel, I computed the average FC across all subjects as a $n \times n$ global covariance matrix and eigendecomposed this FC matrix into its principal components $C = V'EV$. Then, for each state, I randomly chose a set of principal components so that the total explained variance of this set does not go over a certain threshold $\epsilon$. Next, I generated state-specific covariance matrices by permuting the values within each chosen eigenvector. I followed this procedure to generate six different states, each with a different covariance matrix $C_k$. This equals $100 \times 99/2$ parameters per ground-truth state, for a total of 29700 parameters. This was done under two different conditions: for a smaller threshold $\epsilon = 0.1$ and for a larger threshold $\epsilon = 0.2$; i.e., in the $\epsilon = 0.2$ case I permuted more eigenvectors than in the $\epsilon = 0.1$ case, making the states more different to each other, and, therefore, making the subsequent HMM

estimation easier to be performed accurately. Once I had the six ground-truth states, I generated ground-truth state time courses according to a transition probability matrix where the diagonal is 25 times higher than the off-diagonal, so that it is 25 times more probable to remain in the current state than to switch to a different one. Then, I sampled 100 subjects worth of data with 1000 time points each, where the sampling was done according to a Gaussian distribution with covariance given by the active state at each time point. Finally, I estimated HMM-Gauss, HMM-PCA and Mix-PCA models on this data set for a grid of number of states $K$ = 4, 5, 6, 7, 8 and number of principal components $p$ = 5, 10, 20, 30, 40. I repeated the entire process 10 times, calculating, for each model, data set and combination of parameters, (i) the cross-validated likelihood, and (ii) the accuracy with regard to the ground truth. The cross-validated likelihood, used as a quantitative way to assess the models and perform model selection, is sometimes preferred to other methods based on penalised likelihood when the model assumptions are too far from the true generating distribution of the data [31], as it is the case with brain data. Accuracy in this case was defined as how well the estimated state time courses could predict (in a least-squares sense) the ground-truth state time courses.

Fig 4 shows the results of the analysis. The top panels reflect the cross-validated likelihood. In all models, the cross-validated likelihood is able to estimate that at least 6 states are required, but it can hardly distinguish between 6 and higher values of $K$, suggesting that for 7 and 8 states the amount of model overfitting is negligible. For HMM-PCA and Mix-PCA, the cross-validated likelihood favours solutions with larger numbers of PCs. For HMM-Gaussian (which input data is $\boldsymbol{Y}$), we cannot straightforwardly compare different numbers of PCs, and neither can it be compared to HMM-PCA (which input data is $\boldsymbol{D}$), since the likelihood is computed using different data for each choice of $p$ (number of PCs). The middle panels show the error in predicting the ground state time courses, and the bottom panels show that same information as a function of the number of parameters. HMM-Gaussian and HMM-PCA have similar accuracies despite the fact that the ground-truth generating distribution is Gaussian. HMM-PCA outperformed Mix-PCA, specially in the harder $\epsilon = 0.1$ case (right panels), highlighting the importance of the HMM temporal regularisation.

Overall, this section showed that, in simulated data mimicking some aspects of real data, HMM-PCA competes well with, or outperforms, alternative models. These simulations also show that the theoretical benefits of HMM-PCA presented above will apply to a lesser or greater degree depending on the characteristics of the data set under study and the models setting.

**Real data experiments.** Focusing on the HMM-based solutions, I next compared HMM-PCA and HMM-Gaussian on real resting-state fMRI data using 820 subjects from the HCP data set, where each subject underwent four 15-min sessions (TR = 750ms) in the scanner. I used a data-driven parcellation obtained through spatial independent component analysis (ICA) with 50 components; again, see [3] for details about preprocessing and the computation of the ICA time series. The time series of these ICA components were then standardised separately for each session, and then submitted to a stochastic variety of the HMM inference that is specially designed to deal with big volumes of data [16]. Since the estimation of the HMM parameters may return (for the same model and data) slightly different results for each run of the inference [32], I ran the inference five times per model, with $K$ = 12 states and $p$ = 24 principal components each. In brief, in what follows I show that the main FC patterns were not anatomically very different between the two approaches, but that these differences were behaviourally informative. This can be considered as indirect evidence of the superior sensitivity of HMM-PCA.

Asking whether the HMM-PCA states represent meaningful patterns of FC is not straightforward here because there is no ground-truth available. Since both HMM-Gaussian and

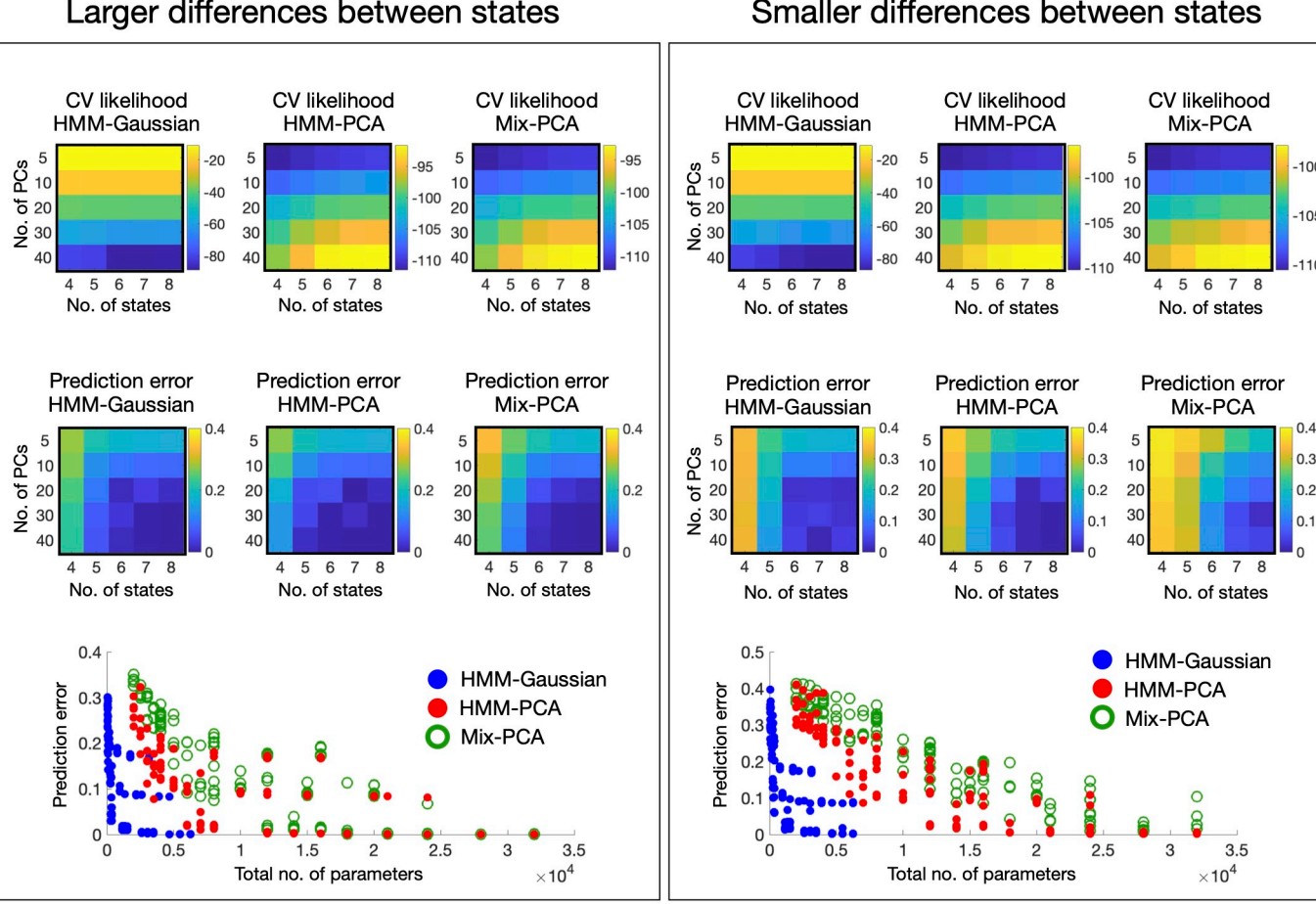

**Fig 4. Empirical comparison of HMM-Gaussian, HMM-PCA and Mix-PCA on simulated data.** Two different conditions were tested: one with larger (A) and the other with smaller differences between states (B). The models were inferred for a range of states $K$ and principal components $p$, and the entire process was repeated 10 times (averages are presented); see main text for details. The top panels show the cross-validated likelihood for each combination of parameters and model. The middle panels show the error of each solution (averaged across repetitions of the experiment) in predicting the ground-truth state time courses. The bottom panels show the errors as a function of model complexity for each model.

HMM-PCA build on PCA, it is however expected that both approaches should be able to capture the main trends in the data to a relatively comparable extent. Given that HMM-Gaussian was shown to produce meaningful estimations in previous work [8, 21], proving that this is the case would situate HMM-PCA on first base. As an example, Fig 5A presents connectedness maps for two given states, where connectedness (or degree) is defined as how much each region correlates with the rest of the brain. The maps were centred across states, such that, if a region exhibits a positive value within a given state, then that region is more correlated to the rest brain's voxels within this state than on average. One of the states is closely associated to the default mode network [2], and the other to the sensorimotor and visual systems. For these two states, both methods capture largely similar anatomical features. Based on the correlation between their FC patterns (specifically, by transforming the states' covariance matrices into correlation matrices, taking the Fisher transformation, and then correlating the off-diagonal elements of these matrices between each pair of states), I then used the Hungarian algorithm [33] to match each HMM-PCA state to a HMM-Gaussian state. On the left, Fig 5B shows the resulting HMM-PCA vs HMM-Gaussian correlations for each pair of states, where the

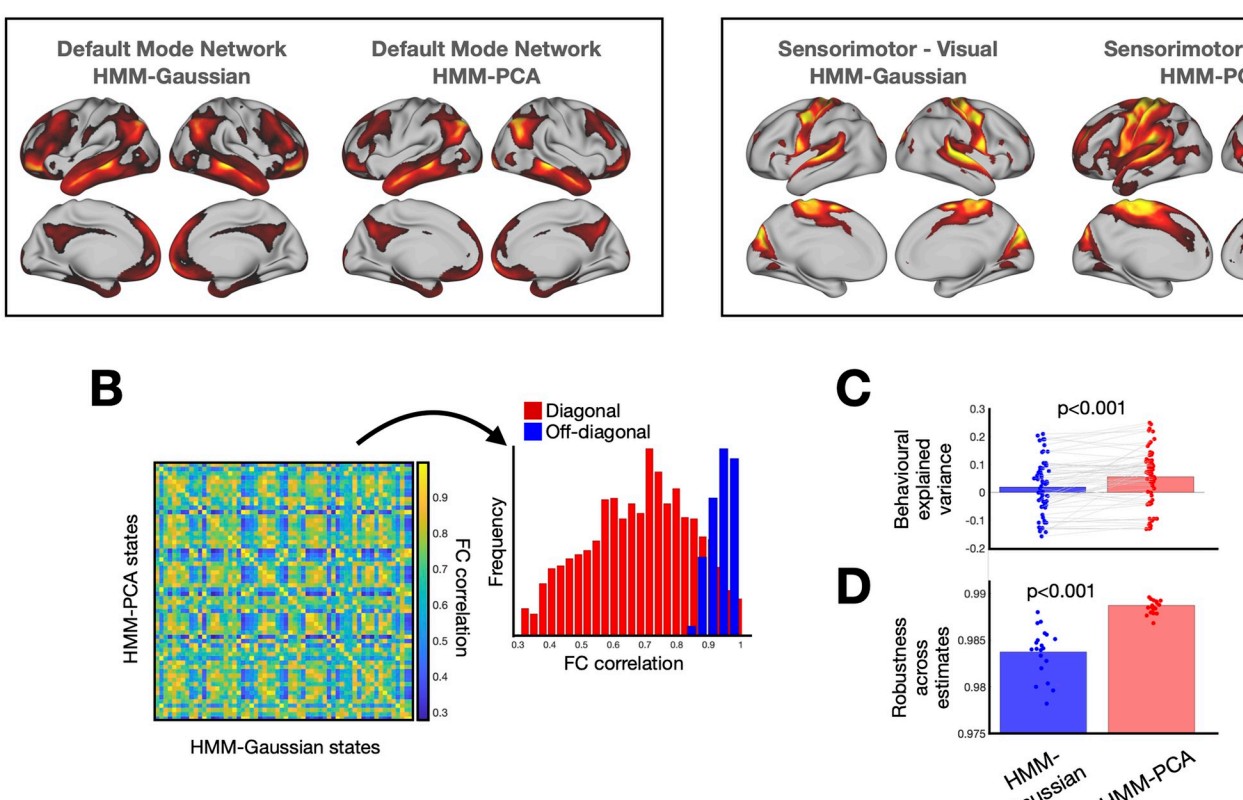

**Fig 5. Comparison of HMM-Gaussian and HMM-PCA on resting-state data from 820 Human Connectome Project subjects. A**. Two example states per model, default mode network and sensorimotor-visual, where the maps reflect the degree, i.e. the total amount of connectivity between each voxel and the rest of the brain. **B**. The states are relatively comparable across the two models. Left: Correlation matrix between the HMM-PCA and the HMM-Gaussian states (in terms of Pearson's correlation between the off-diagonal elements of the states' FC matrices) across 5 runs of the inference. Right: Distribution of the diagonal elements (red) and off-diagonal (blue) elements, where the red elements reflect the similarity between corresponding states (i.e. an HMM-PCA state and an HMM-Gaussian state that correspond to each other), and the blue elements reflect the similarity between non-corresponding states. **C**. HMM-PCA produces solutions that are more explanatory of behaviour. Using the fractional occupancy across runs (i.e. the amount of time spent on each state), the HMM-PCA models are better able to predict behavioural traits across a range of demographical, intelligence, personality and affective-related variables; each dot represent the explained variance (in terms of Pearson's correlation) for the cross-validated prediction of one behavioural trait, and the bars represent the average across traits. **D**. HMM-PCA produces solutions that are more robust than HMM-Gaussian across repetitions of the inference; robustness is here measured as the capacity to predict (using cross-validation) the state time courses of one HMM estimate using the state time courses of another HMM estimate.

diagonal corresponds to states that were matched, and the off-diagonal to any other pair of states. On the right, Fig 5B shows the distribution of between-state correlations for states that were matched to each other (blue) and, to provide context, the correlations between states that were not matched (red). This indicates that the main time-varying FC patterns are relatively well preserved between the two methods.

Then, I sought to investigate whether the theoretical benefits of HMM-PCA have indeed a practical impact on real data. First, for each run of the inference, I extracted the fractional occupancies, defined as the percentage of time spent on each state for each session. I then used these $12 \times 5$ fractional occupancy values as features to predict a collection of behavioural traits. In particular, I chose 63 traits across different domains including demographical, affective, personality- and intelligence-related [5], and performed cross-validated predictions respecting the family structure of the HCP data [34]. As shown in Fig 5C, the cross-validated predictions were found to be significantly more accurate for HMM-PCA than for HMM-Gaussian

(p-value = 0.001, permutation testing). In accordance with previous work [35, 36] the accuracies in predicting HCP traits using FC features are relatively modest, with the average explained variance being lowered by traits that are particularly hard to predict (e.g. the personality traits). The average is, however, significantly higher than zero. Note the grey lines connecting each trait's prediction between the two models reflecting that the two models tend to perform better on the same traits. Furthermore, as shown in Fig 5D, the HMM-PCA solutions were also more robust across runs of the HMM inference (p-value = 0.001, permutation testing); robustness in this case was measured as the capacity to predict, through cross-validation, the state time courses of a given HMM estimate using the state time courses of another HMM estimate (for a total of 20 pairs of estimates, having estimated the models five times).

These results were obtained running the HMM on a single hyperparameter configuration ($K$ = 12 states and $p$ = 24 principal components), so it is possible that the edge exhibited by HMM-PCA over HMM-Gaussian does not generalise to other configurations. To test this, I estimated the models on a grid of hyperparameters, $K$ = 4, 5, 6, 7, 8 and $p$ = 5, 10, 20, 30, 40, and reran the predictions on the behavioural traits for each pair of hyperparameters. Fig 6 shows the results of comparing the HMM-Gaussian vs the HMM-PCA predictions, where I performed (permutation-based) statistical testing in both directions: is HMM-PCA better than HMM-Gaussian at predicting behaviour? and, is HMM-Gaussian better than HMM-PCA at predicting behaviour? The left panels show a p-value for each combination of hyperparameters, and the right panels show histograms of p-values across all configurations. Finally, I used the nonparametric combination algorithm [32, 37], to combine the different tests across all combinations of parameters into a single, aggregated p-value. This procedure showed that HMM-PCA was significantly better at predicting behaviour than HMM-Gaussian for the considered grid of hyperparameters.

In summary, these results suggest, albeit in an indirect way, that HMM-PCA might be better able to describe time-varying FC in high-dimensional fMRI data, and that the theoretical limitations discussed above can in fact have an impact in practice.

## Discussion

By fusing dimensionality reduction and time-varying FC in one single model, the presented HMM-PCA approach can bypass some important limitations of the common two-step procedure, where dimensionality reduction and time-varying FC estimation are performed in sequence. This was shown on simulations and on real fMRI data, where the HMM-PCA states were significantly better able to predict a number of behavioural traits. Importantly, the cause of the reported biases does not inherently lie on the high dimensionality of the data, but on the use of PCA as a way to reduce such dimensionality. Therefore, although for practical reasons the examples and simulations used throughout the paper are not high-dimensional by normal standards [38], the issues addressed by these examples apply generally.

An important question about the application of these models to real fMRI data is how to decide which model is objectively better. In Fig 3, for example, I showed that the use of PCA changes the estimation from what it would be obtained without using PCA. This is a mathematical fact, but does it mean that the PCA-related prediction is worse? This question depends on what is meant by the goodness of the model. In many practical applications, goodness would be given by some combination of accuracy (here, data likelihood) and model complexity [19]. Here, I used the cross-validated likelihood to assess the models, which can be more appropriate than methods based on the penalised likelihood (like the free energy) when the assumptions of the compared models do not meet the realities of the true generating distribution [31]. Most importantly, although the HMM has modelling assumptions, these do not

### Is HMM-PCA better than HMM-Gaussian at predicting behaviour?

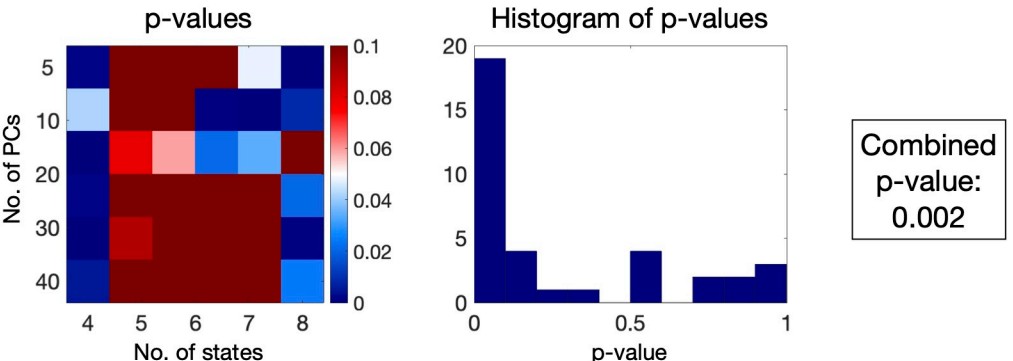

### Is HMM-Gaussian better than HMM-PCA at predicting behaviour?

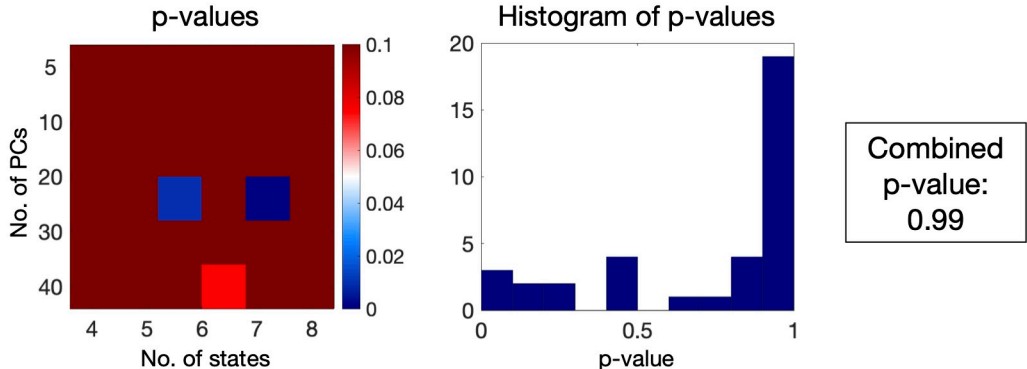

**Fig 6. Behavioural predictions of HMM-Gaussian and HMM-PCA across a range of parameters.** Similarly to Fig 5, the fractional occupancy was used to predict a set of behavioural traits. Permutation testing was used to test whether, for each combination of parameters, the HMM-PCA solutions were better than the HMM-Gaussian solutions (top), and whether the HMM-Gaussian solutions were better than the HMM-PCA solutions (bottom). On the left side: the corresponding p-values, where colder colours mean statistically significant; at the center: histograms of p-values across all combinations of hyperparameters; on the right side: the result of combining all the tests into one single, aggregated test, using the non-parametric combination algorithm –as used in [32].

imply a statement about the underlying biology. For example, using eight states does not necessitate or imply the assumption that there are eight biological states in the human brain (or eight attractors in the system). For this reason, we cannot straightforwardly say that a model is more biologically plausible because it has a higher cross-validated likelihood. Still, a valid theoretical argument can be made: in the asymptotic limit of having infinite data and assuming an ideal parameter inference (i.e. such that overfitting is not a factor), any transformation (like PCA) that changes the estimation from the one obtained from the original (non-PCA) data is undesirable. From Fig 3, it follows that a PCA transformation can exert such detrimental bias, because PCA alters the estimation independently of the amount of data. The extent to which this actually happens in real fMRI data will depend on the data and the preprocessing. For instance, the bias highlighted in Fig 3 would be less severe in data where global signal regression [39] has been performed; otherwise, a single component capturing the global signal could explain a large amount of variance, therefore increasing the *PCA concentration* measure and therefore the distortion. A more complete description of which data sets would

be more or less affected by these issues would possibly necessitate the use of simulations that are more biophysically realistic [40].

Another important question is what these models are actually expressing. Because PCA is computed on the full data covariance, it is theoretically possible for the states transitions to be driven only by changes in the pattern of relative variance across channels with little or no contribution of the between-regions covariance. Here, however, it could be observed that the covariance between regions (above and beyond the variance) is clearly different between states. Otherwise we would not see any meaningful patterns after normalising by the variance (i.e. after transforming the covariance matrices into correlation matrices). The connectivity maps shown in Fig 5A, as well as the between-state FC correlations shown in Fig 5B, support that there are within-session FC modulations driving the estimation. This is in agreement with our previous work on the HCP data, where we showed (i) that HMM states have unique information in the off-diagonal elements of the covariance matrix, and (ii) that HMM models based solely on the variance (i.e. with no covariance) did not predict behaviour as accurately as those that model the full covariance [5, 8].

A limitation of these models is that they might not be suitable to be run in very high-dimensional data, such as the original surface-space of the HCP data (around 90k vertices/voxels). On top an excessive computational cost, without having appropriate additional mechanisms to control the high-dimensionality of the parameter space, overfitting might occur due to the large number of parameters in each state. Indeed, if the data dimensionality $n$ is very high, HMM-PCA (with $n \times p$ parameters per state) might potentially overfit more than HMM-Gaussian (with $p \times (p - 1)/2$ parameters per state), overshadowing the advantages discussed in this paper. Altogether, the presented model and the considered alternatives are, at present, better suited to be run in intermediate spaces, such as those produced by ICA or an anatomical parcellation, than on raw whole-brain space. Of course, each of these choices entail its own biases. An efficient application of prior distributions on the state-specific PCA weights $W_k$, for example promoting sparsity [19, 41], is a promising avenue to help in this direction. Other limitations, such as the incapacity to model long-term temporal dependencies or the fact that the model does not allow for overlapping states, are however not specific of HMM-PCA, but more generally of the HMM framework.

## Conclusion

In this paper, I have addressed the question of how to estimate patterns of time-varying FC in fMRI data. I have shown that the standard approach of sequentially applying PCA and then feeding the resulting PCs to an HMM or mixture model, although useful in practice, may suffer from biases and loss of sensitivity. On these grounds, I have introduced a new variety of the HMM, namely HMM-PCA, where each state is a probabilistic PCA decomposition. Critically, the HMM states not only express a (linearly optimal) dimensionality-reduction of the data, but also encode a correlation pattern between regions.

## Supporting information

**S1 Fig. Graphical representations of the four models.**
(TIFF)

## Acknowledgments

I thank Angus Stevner, Christine Ahrens, Piergiorgio Salvan, Mark Woolrich and Steve Smith for conversations that motivated this work.

## Author Contributions

**Conceptualization:** Diego Vidaurre.

**Formal analysis:** Diego Vidaurre.

**Funding acquisition:** Diego Vidaurre.

**Investigation:** Diego Vidaurre.

**Methodology:** Diego Vidaurre.

**Project administration:** Diego Vidaurre.

**Resources:** Diego Vidaurre.

**Software:** Diego Vidaurre.

**Supervision:** Diego Vidaurre.

**Validation:** Diego Vidaurre.

**Visualization:** Diego Vidaurre.

**Writing – original draft:** Diego Vidaurre.

**Writing – review & editing:** Diego Vidaurre.

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
