## [Decision Letter · Decision Letter 0]

6 Jan 2021

Dear Dr Vidaurre,

Thank you very much for submitting your manuscript "The statistical challenge of finding spontaneous changes in functional connectivity in high-dimensional fMRI data" for consideration at PLOS Computational Biology.

As with all papers reviewed by the journal, your manuscript was reviewed by members of the editorial board and by several independent reviewers.

Your paper was overall very well received. Some aspects need to be better clarified and inserted in the state of the art. In light of the reviews (below this email), we would like to invite the resubmission of a significantly-revised version that takes into account the reviewers' comments.

We cannot make any decision about publication until we have seen the revised manuscript and your response to the reviewers' comments. Your revised manuscript is also likely to be sent to reviewers for further evaluation.

Sincerely,

Daniele Marinazzo

Deputy Editor

PLOS Computational Biology

Reviewer's Responses to Questions

**Comments to the Authors:**

Reviewer #1: Summary:

In this paper, the author proposes an HMM-PCA model to explain time-varying FC in resting state data. The idea is very nice and potentially very useful, but the methodology and results do not adequately support the claims. The paper needs to be improved.

General feedback:

To motivate this approach, I think the author needs to clearly show the generative model (in form of a graphical model) and also report the model evidence, when performing model inversion. As for the inference technique, if one commits to maximum likelihood, then a grid search over the number of HMM hidden states and the number of PCA latent states is essential. Alternatively, if one turns to Bayesian PCA (reference 19), then Bishop's hyperparameter approach will determine the effective dimensionality of PCA, but the optimal cardinality of HMM’s latent space would still rely on grid search. Either way, the model evidence (accuracy minus complexity) will determine whether this new model is superior to HMM+PCA (in 2 steps) or Gaussian mixture model, in the present application. Ideally, this superiority needs to be demonstrated on high-dimensional (simulated) data first, because the claims pertain to the curse of dimensionality. In short, face validation of a new model is conventionally established based on model comparison (in terms of model evidence) and accuracy of the posteriors (with respect to the ground truth). If there is any such thing as bias/distortion or loss of sensitivity (as the author suggests) due to the two-step procedure, these will also show up in the results, alongside the more important model evidence.

Detailed comments:

- Speaking of the issue of high dimensional data, please note that 50-100 regions/ICs is not really high dimensional when the time series are concatenated across subjects. Moreover, your simulations are low-dimensional (with 10 regions/ICs). So you are not actually tackling the high-dimensional scenario (of many voxels and few time points) neither in the simulations nor in empirical data, to show the reader the gravity of this problem and how you have solved it.

- page 2, 2nd paragraph: this criticism to the sliding window analysis (that mere fluctuations may be mistaken for non-stationarity in the connectivity dynamics) applies to the hidden Markov model as well, unless model evidence rules out the existence of only one underlying connectivity state.

- page 3, 1st paragraph: Please rephrase. Using ambiguous statements such as "works reasonably well in practice for many (but not all) fMRI data" without bringing in quantitative support or a credible reference is not appropriate. Or, for instance, saying that a method "is not designed for the final goal of the analysis" without specifying that final goal, is not precise enough for a scientific text.

- page 3, 2nd paragraph: I can't see why the FC matrices are being interpreted as instantaneous, when in practice they are the covariance matrices of the Gaussian distributed observations (of each state).

- page 3, 4th paragraph: principal components are by design orthogonal. I don't see your point about transient departure from orthogonality. Please elaborate if this is actually what you meant.

- page 3, section 2.2: when a new probabilistic model is proposed, it is common to show the (generative) model structure as a graphical model, clearly showing the conditional (in)dependencies, and the model parameters and hyperparameters.

- page 4, 1st paragraph: It seems that you are using probabilistic PCA, not Bayesian PCA (with prior over W), as per your reference 19 (Bishop, Bayesian PCA).

- page 4, below Eq.1: I think discounting the mean needs further motivation. Does it have computational or practical advantages? More importantly, do you get higher model evidence by fixing all the means at a prior value of zero?

- page 4, middle of page: "Note that the estimation of u_tk only depends on the data at time point t and on pi_k, ignoring the data temporal structure". It is true that temporal order is not accounted for in mixture models; however the responsibilities depend on all the data points through the model parameters. So each u_tk does not depend *only on time point t and on pi_k*. Please rephrase to avoid misunderstanding.

- page 4, middle of page: "In the HMM-PCA case, we instead have the state latent variables xk modelled as an order-1 Markovian process...". Please draw the generative model to show how the HMM states map to the PPCA states, which are then combined with W to generate the data dt. Also please make sure the reader realizes the difference between these two types of latent states; i.e. the (categorical) hidden states of HMM, as opposed to the (Gaussian) latent states of PPCA. It is useful to stress their dimensionalities as well (K versus p0).

- page 4, Eq.3: Complete data log-likelihood (LHS) is also a function of the model parameters.

- page 4, Eq.4: this is not the familiar form of Bishop's PPCA. If it comes from another source, please provide the reference. Otherwise please put your derivation in the appendix.

- page 6: This whole bias/distortion story of PCA (assuming it is serious) has not been demonstrated well. Note that in Fig 2B most of the computed distortion is below 2%. Plus, how often does the first PC become so dominant in a FC pattern to take up 86% of the variance? I think to motivate your approach, you can compare model evidence of HMM+PCA (as a two-step procedure) to your proposed HMM-PCA. Even Fig 2A is not convincing, because there is no ground truth here, so we can not say PCA has caused any distortion. HMM+PCA is simply suggesting a different number (and sequence) of hidden states for the HMM, which may actually be more plausible than the results on the original (unrotated) data, in terms of model evidence. Note that you have not shown model comparison results for the number of hidden states, neither on the original data, nor on the PCA-rotated data. So I think this exemplar plot is not sufficiently supporting the distortion/bias claim.

- page 6, section 2.3.2: the same problem (as above) exists when discussing loss of sensitivity. Although this claim comes with simulated data, still the dimensionality is low (10 regions) and there is no model comparison result for different number of HMM hidden states and different number of PCA components. Accuracy of the state sequence through time is not adequate support for superiority of one method over another, when the correct number of latent states have been provided and no model evidence is reported. And even these accuracy values have not been statistically compared. Please reconsider these simulations.

- page 8, section 3: Same problem. Maximum fractional occupancy and behavioral relevance can not replace model evidence. And please note that the behavioral relevance is barely different from zero. Perhaps if you compute the effect size alongside the p-value it becomes clearer that this may not be a very strong argument in favor of your model.

This model has great potential and it deserves to be presented and motivated accordingly. Best of luck.

Reviewer #2: This article concerns the statistical difficulties in estimating time-varying functional connectivity in the brain from fMRI images where spatial dimensionality is high but the temporal windows for the time-resolved networks are small. The authors show that conventional time-averaged PCA methods can exhibit bias towards the least relevant components because all are given equal weight. They propose a hybrid method which uses (i) PCA to reduce the data at each time sample, and (ii) HMM to identify changes in the state of the FC network. The proposed method is tested on simulated and real fMRI data.

The paper is very well written overall. I especially appreciated the Introduction as I am not familiar with these statistical methods. Consequently my review is rather general in nature. I garnered that the proposed method improved the statistical result by using a single-step procedure to simultaneously reduce the data and estimate the functional connectivity at each state. However it was not clear to me how the algorithm worked. Specifically, I did not understand where the K different state-specific covariance matrices originated. Do they come out of the analysis?  Is the number K something that must be chosen beforehand? Perhaps the author can clarify these basic issues for readers who are not familiar with Mix-PCA.

The layout of the manuscript could be improved. It is not clear where the Methods end and the Results begin. Methods sections 2.3 onwards would be better placed under Results.

The results themselves are well-presented and I have no issues there.

Section 2.3.1 provides a good explanation of why the lower-order components in time-averaged PCA introduce biases. The third paragraph "When we apply PCA..." summarizes it particularly well.

Section 2.3.2 provides nice results for the better sensitivity of the HMM-PCA method.

Section 3 compares the methods on real fMRI data and finds better results for the proposed method. The author admits (page 9 top) that is difficult to interpret the results in the absence of a ground truth in real fMRI data. Might it be possible to use a generative model of fMRI to test the algorithms in a scenario where the ground truth is known? Perhaps the author can comment on this in the Discussion. 

Upon finishing the paper, I felt that it could have done more to make the proposed method available to the reader as a software package. At very least I encourage the author to make his existing code available as a demonstration. Otherwise the reader is obliged to transform the mathematical equations into their own software, which can be risky and time-consuming.

Minor points

----------

Section 2.1, second paragraph. "... as zero mean Gaussian distributions."

Gaussian distributions of what? I didn't understand this statement.

Section 2.1, third paragraph. "... transient departures from orthogonality."

   I was confused what you meant by this. Do you mean that the principle components of one state differ from those of the next state? Can you please clarify?

Section 2.2, first paragraph.

    (1) Typo: "Various [of the] elements [of] this model are ...."

    (2) How is K determined? Please elaborate.

Section 2.2, second paragraph. "the Mix-PCA model uses K different PCA projections"

    Again, I wasn't clear to me where these K different projections came from.

Section 2.2, second paragraph. "... using the EM algorithm"

    What does "EM" mean? Perhaps I missed it but I don't think it was ever explained.

Section 2.2, third paragraph.  "...using the forward-backward equations ...."

    What are the 'forward-backward' equations? Are these a standard part of the Mix-PCA method?

Section 3, Title: 'Results on real fMRI data'

    This title is a little misleading as it suggest that the previous results (section2) are not based on real fMRI data. Whereas Fig 2 is based on real fMRI data, albeit manipulated data. Likewise for Fig 3, Scenario 2.

Reviewer #3: In this paper the author has suggested a new approach in the HMM family that tries to combine PCA with HMM in a more unified way. The aim of this approach is to remedy the issue of high dimensionality of fMRI data. I find the method quite interesting and can think of many instances where such an approach would be quite useful. My major issue with this paper is that it seems a little rushed (for the lack of better term) in both introduction and discussion. I think if the authors add some more information to both introduction and discussion, the paper would be improved greatly.

Specific comments:

1. First and foremost, I do not find the title of the paper representing the core of this paper. This paper does not discuss the challenge of finding spontaneous changes in functional connectivity” a lot. Instead, a new approach is proposed that might help solve some of those “challenges”. When I read the title, I thought this paper is a review/commentary paper.

2. I also find the introduction overly short. The authors have only given sliding window Pearson correlation (SWPC) as a different category of methods (the rest are all HMM). First although SWPC is one the most know method in this field, there are some more recent connectivity estimation that have been proposed recently. For example, shared trajectory by (Faghiri et al. 2020) or instantaneous phase synchrony (Pedersen et al. 2018) and even Multiplication of Temporal Derivatives (Shine et al. 2015). As these are more instantaneous estimators (compared to SWPC where the window size makes sure the estimand are not instantaneous) I find them more relevant to the current work and therefore should probably be cited too. Second, I don’t think these estimators should be compared to HMM at all. As HMM tries to directly estimates the FC states whereas these estimators aim to estimate connectivity and an additional step is usually required to estimate FC states from these connectivity time series. For example, one can use kmeans to estimate the FC states from connectivity time series (Allen et al. 2014). Right now if someone reads the introduction it seems you are comparing SWPC with HMM family which is not quite accurate.

3. In the “Loss of sensitivity” section the author states “for example, this would correspond to a data set with sampling rate or TR≈1.1s, and 15min worth of data per session.” I do not understand how the authors has calculated the TR. Based on what I understand no sampling/resampling has been done to get a sampling rate value. This simulation can be from any other TR value too. And why is this information is given at all? The authors do not use it any other place to derive any kind of result.

4. Is it possible that the stability of the other HMM variants (gaussian HMM) is lower compared to the HMM PCA? i.e. can gaussian HMM predict better if it is run more than 5 times?

5. I do not see any mention of pre-processing of the fMRI data. Even if the authors have not done this at least a short summary would be needed (or a citation of where to find that information).

6. I find the discussion of this paper quite underwhelming. I think the author need to expand this part too. for example, a little more on the limitation of the proposed method? I see that the authors mention that their method is sensitive to the temporal ordering of the time series (which is true) but it has a limit. If I am understanding it correct it only care about temporal ordering for one time point. This can be considered a limitation as I believe this method would be unable to see slower changes. This point to another limitation of this study. The proposed method only find hard changes in the state and not the more fuzzy changes (at least the simulation is designed like that).

References:

• Faghiri, A., Iraji, A., Damaraju, E., Belger, A., Ford, J., Mathalon, D., ... & Turner, J. (2020). Weighted average of shared trajectory: A new estimator for dynamic functional connectivity efficiently estimates both rapid and slow changes over time. Journal of neuroscience methods, 334, 108600.

• Pedersen, M., Omidvarnia, A., Zalesky, A., & Jackson, G. D. (2018). On the relationship between instantaneous phase synchrony and correlation-based sliding windows for time-resolved fMRI connectivity analysis. Neuroimage, 181, 85-94.

• Shine, J. M., Koyejo, O., Bell, P. T., Gorgolewski, K. J., Gilat, M., & Poldrack, R. A. (2015). Estimation of dynamic functional connectivity using Multiplication of Temporal Derivatives. NeuroImage, 122, 399-407.

• Allen, E. A., Damaraju, E., Plis, S. M., Erhardt, E. B., Eichele, T., & Calhoun, V. D. (2014). Tracking whole-brain connectivity dynamics in the resting state. Cerebral cortex, 24(3), 663-676.

**Have all data underlying the figures and results presented in the manuscript been provided?**

Reviewer #1: Yes

Reviewer #2: Yes

Reviewer #3: Yes

PLOS authors have the option to publish the peer review history of their article (what does this mean?). If published, this will include your full peer review and any attached files.

Reviewer #1: No

Reviewer #2: No

Reviewer #3: **Yes: **Ashkan Faghiri
---

## [Decision Letter · Decision Letter 1]

28 Feb 2021

Dear Dr Vidaurre,

Thank you very much for submitting your manuscript "Fusing dimensionality-reduction and time-varying functional connectivity estimation in a single model" for consideration at PLOS Computational Biology.

While we appreciate the way you addressed some issues, we feel that some important issues, as identified by reviewer 1, are still present, and to some extent more evident after this round of revision.

Some of these concerns are fundamental, and as such they could be unsurmontable. Still I think that it would be good to go through them.

This field is rapidly advancing, and we appreciate that you are providing a robust framework with modern tools. OIn the other hand it should be clear which aspects are novel with respect to previuous work (even beyond neuroimaging data analysis), and how the results can be interpreted.

We cannot make any decision about publication until we have seen the revised manuscript and your response to the reviewers' comments. Your revised manuscript is also likely to be sent to reviewers for further evaluation.

Sincerely,

Daniele Marinazzo

Deputy Editor

PLOS Computational Biology

Daniele Marinazzo

Deputy Editor

PLOS Computational Biology

Reviewer's Responses to Questions

**Comments to the Authors:**

Reviewer #1: I appreciate that the manuscript has been improved with this revision. Performing more simulations, showing graphical models, computing model evidence, plus a number of other clarifications have all been valuable additions. I agree that HMM-PPCA is a potentially useful addition to the field of dynamic FC analysis.

However, there are still fundamental issues that we disagree about. In the following, I will elaborate on these points.

• Introduction/motivation

To start with, please remember to cite the HMM-mixture-of-Bayesian-PCA model of Mauricio Alvarez and Ricardo Henao (2007) as prior work. Actually your HMM-PPCA model seems like a special case of their formulation.

From an organization point of view, the motivation section (called limitations of two-step approach) should go under Introduction, and not Results. But more importantly, I don’t think the motivation content is appropriate.

Specifically, I still can not sympathize with the distortion story of PCA, used to motivate your model. You are saying that two-step PCA+HMM produces results different from HMM on raw data, when many PCs are included. Lets forget for now that you showed us this distortion is about 1% for 84% explained variance. I repeat what I mentioned in the previous review: I don’t understand why anyone would expect PCA+HMM results to be identical to HMM. PCA is a data transformation/reduction technique. Of course it will discard certain details to facilitate discovery of the underlying structure in the data. This is not distortion, it’s the natural compromise of data reduction.

If you still see this as an important problem, have you at least shown that HMM-PPCA solves it? Have you shown that HMM-PPCA results are very similar to HMM on raw data? I didn’t find any such results. When you stress an issue as the motivation of your work, the reader is waiting to see how you have resolved this issue.

I understand you need to motivate the model. Why not use all the motivations in the mixPPCA/mixFA literature? For example: the benefits of local linear sub-models, having fewer parameters to estimate than a full-covariance Gaussian mixture model, not having to impose axis aligned covariance matrices to reduce the number of parameters, all the advantages of probabilistic PCA over classic PCA, the temporal structure modeled with HMM, etc. I am sure you know the following references, so you can easily get inspirations from their introductions: Bishop and Tipping (1999), Bishop and Winn (2000), Ghahramani and Beal (1999).

If you have been thinking that global PCA prior to HMM is not as efficient as local PCA (using mix-PPCA or your extension as HMM-PPCA) in capturing and preserving local structure, I totally agree. But this is not what you have explained for the reader.

o Simulation Design

Simulations should be designed to showcase the important aspects of a proposed model. In the new high-dimensional simulations of the manuscript, the (effective) dimensionality of the data, per component/state, is not specified. This is important because you need to show that HMM-PPCA can recover both the number of HMM latent states and the intrinsic dimensionality of each state-specific PPCA. Actually your lower-dimensional simulations had this aspect encoded in the number of retained eigenvectors of the covariance, but the new simulations miss this point. When we don’t know the dimensionality of the principal subspace in the simulations, we can not judge the model inversion and comparison results.

o Coupled update equations

(Expected) complete data log-likelihood (Eq. 4) should be written for the whole HMM-PPCA model (not per state), in which case the EM update equations of PPCA and HMM become coupled. Intuitively, this is because the likelihood term (used to update the HMM state) now depends on PPCA posteriors as well. This is an important point to show, as you are offering a unified model. Check out the following references for similar derivations of coupled update equations:

- Bishop and Tipping's (1999) mix-PPCA (Appendix C, Eq. 73)

- Bishop and Winn’s (2000) variational treatment of mix-BPCA (Section 3.1)

- Alvarez and Henao’s (2007) derivations for the HMM-mix-BPCA model (Appendix B)

o Latent space identification

My concerns here can be best summarized as a question: How does the author intend to identify the optimal number of latent states for HMM and PPCA on empirical data? There is no answer in the manuscript.

There are standard approaches in the literature for identifying the number of latent states in such models: grid search, cross-validation, and Bayesian treatment (with hyper-priors over the loading matrix). Check out for example: (Bishop and Winn, 2000; Bishop and Tipping, 1999; Alvarez and Henao, 2007; Ghahramani and Beal, 1999) to see how the latent space dimensionality is specified and then recovered successfully. While the fully Bayesian treatment has undeniable advantages, all three solutions are valid. My point is that this problem has been solved in the literature, and your application is no exception.

Also note that the dimensionality of the principal subspace can differ across states/components in the HMM-PPCA model. In your application this means that different functional states can express their intrinsic complexities through the dimensionality of their respective principal subspaces.

So, on the one hand, optimizing the latent space is essential for model fitting on a given dataset. On the other hand, this step is important for between-group analysis, e.g. between a normal and patient group. Once the model has been fitted to each dataset separately, one may find out that the functional components from one group fit into lower dimensional subspaces, or may need fewer HMM states to be explained, than the other group, which can be nicely interpreted. If latent variable dimensionalities have not been optimized on each dataset, such between-group analysis would not be feasible.

I can see that you tried the grid search approach, and computed free energy as a proxy for model evidence. This is a valuable step in the right direction, because free energy computation can guide the choice of model parameters and also facilitate between-model comparison (e.g. HMM-PPCA vs. mix-PPCA). Apparently you think free energy has failed you in model comparison. I discuss this next.

• Model comparison with free energy

Please note that what I call free energy (F) is the negative of free energy in physics, or ELBO in machine learning. There could be a number of reasons why free energy is not guiding your model comparison to optimize the latent space dimensionality and to compare competing models of different structure:

1) I suspect that you have not computed free energy for the whole hierarchical model. I had a glance at your code, and F seems to pertain to the HMM part of the model only. I can’t see the complexity terms neither for the PPCA nor for the transition matrix. If that is indeed the case, it explains why F is not penalizing higher complexity and keeps encouraging more and more dimensions/states (similar to over-fitting of maximum likelihood schemes). You may want to have a look at the Appendix of Friston et al. (2015) to see how the free energy of a hierarchy is written in terms of the expected free energy of the lower level minus the complexity of the higher level.

2) A second problem might be the way you are parametrizing the likelihood. I remember I mentioned in the previous review that taking out the ‘mean’ is not justified. That is because when you fix all the means to zero, it’s like trying to disentangle concentric ellipsoids. You may want to put the mean back and see whether the state cardinality shows itself.

3) A third solution is adopting a fully Bayesian treatment (with prior over PPCA loadings). At least for PPCA dimensionality discovery this approach has proven very successful in the literature.

• Replacements for model evidence

The author seems to propose fractional occupancy of states and behavioral relevance as model evaluation criteria. As I explained before, these can not replace model evidence. (As a side note: one can use other approximations to model evidence, other than F, or even use sampling to compute model evidence).

Speaking of fractional occupancy, I have seen in the literature that a balanced fractional occupancy of states is more probable in healthy brains, than pathologic ones. However, fractional occupancy is not the cost function for optimizing the parameters of your model. This sort of domain-specific information can be encoded as ‘prior’ belief, e.g. on the form of the transition matrix. As for behavioral relevance, only once the model optimization is over can one inspect relation to behavior. Comparing behavioral relevance on (structurally different) models with arbitrarily chosen parameters is not meaningful, to claim superiority of one model over another.

I think taking care of these points would go beyond a standard revision. I would encourage the author to go through the reviews, repeat some of the analyses, and resubmit a fresh version of this work. I still think the idea is great and should definitely be added to the literature soon. Best wishes.

• References:

Alvarez, M. and Henao, R., 2007. Hidden Markov Bayesian Principal Component Análisis. Lecture Notes in Computer Science, Neural Information Processing/ICONIP.

Bishop, C.M. and Winn, J.M., 2000, June. Non-linear Bayesian image modelling. In European Conference on Computer Vision (pp. 3-17). Springer, Berlin, Heidelberg.

Tipping, M.E. and Bishop, C.M., 1999. Mixtures of probabilistic principal component analyzers. Neural computation, 11(2), pp.443-482.

Ghahramani, Z. and Beal, M.J., 1999, December. Variational Inference for Bayesian Mixtures of Factor Analysers. In NIPS (Vol. 12, pp. 449-455).

Friston, K., Zeidman, P. and Litvak, V., 2015. Empirical Bayes for DCM: a group inversion scheme. Frontiers in systems neuroscience, 9, p.164.

Reviewer #2: The authors have addressed all of my concerns.

Reviewer #3: I think the author have answered all my concerns. the paper is in a good state to be published.

**Have all data underlying the figures and results presented in the manuscript been provided?**

Reviewer #1: Yes

Reviewer #2: Yes

Reviewer #3: Yes

PLOS authors have the option to publish the peer review history of their article (what does this mean?). If published, this will include your full peer review and any attached files.

Reviewer #1: No

Reviewer #2: No

Reviewer #3: **Yes: **Ashkan Faghiri
---

## [Decision Letter · Decision Letter 2]

31 Mar 2021

Dear Dr Vidaurre,

We are pleased to inform you that your manuscript 'Dimensionality reduction and time-varying functional connectivity estimation in one single model' has been provisionally accepted for publication in PLOS Computational Biology.

When you submit the final version, please fix the typo pointed out by the reviewer.

Best regards,

Daniele Marinazzo

Deputy Editor

PLOS Computational Biology

Daniele Marinazzo

Deputy Editor

PLOS Computational Biology

Reviewer's Responses to Questions

**Comments to the Authors:**

Reviewer #1: I would like to thank the author for his scientific attitude in taking the reviews. I think switching to cross-validation was a wise decision for a maximum-likelihood scheme. The other revisions have also improved the paper and made it clearer.

Just a typo: Eq (4) last line: <log p=""> is right, (x_0, not x_t)

I recommend this version for publication. Best wishes to the author and congratulations.</log>

**Have all data underlying the figures and results presented in the manuscript been provided?**

Reviewer #1: Yes

PLOS authors have the option to publish the peer review history of their article (what does this mean?). If published, this will include your full peer review and any attached files.

Reviewer #1: No

---

## [Editor Report · Acceptance letter]

9 Apr 2021

PCOMPBIOL-D-20-02157R2 

Dimensionality reduction and time-varying functional connectivity estimation in one single model

Dear Dr Vidaurre,

I am pleased to inform you that your manuscript has been formally accepted for publication in PLOS Computational Biology. Your manuscript is now with our production department and you will be notified of the publication date in due course.

With kind regards,

Katalin Szabo
